# Empagliflozin Ameliorates Type 2 Diabetes-Induced Ultrastructural Remodeling of the Neurovascular Unit and Neuroglia in the Female *db*/*db* Mouse

**DOI:** 10.3390/brainsci9030057

**Published:** 2019-03-07

**Authors:** Melvin R. Hayden, DeAna G. Grant, Annayya R. Aroor, Vincent G. DeMarco

**Affiliations:** 1Diabetes and Cardiovascular Center, School of Medicine, University of Missouri, Columbia, MO 65212, USA; aroora@health.missouri.edu (A.R.A.); demarcov@missouri.edu (V.G.D.); 2Division of Endocrinology and Metabolism, Department of Medicine, University of Missouri, Columbia, MO 65212, USA; 3Electron Microscopy Core Facility, University of Missouri, Columbia, MO 65212, USA; GrantDe@missouri.edu; 4Research Service, Harry S. Truman Memorial Veterans Hospital, Columbia, MO 65201, USA; 5Department of Medical Pharmacology and Physiology, University of Missouri, Columbia, MO 65212, USA

**Keywords:** astrocyte, endothelial cell, microglia, mitochondria, myelin, neuroglia, oligodendrocyte, pericyte, sodium glucose co-transporter 2 inhibitor (SGLT2i), white matter

## Abstract

Type 2 diabetes is associated with diabetic cognopathy. Anti-hyperglycemic sodium glucose transporter 2 (SGLT2) inhibitors have shown promise in reducing cognitive impairment in mice with type 2 diabetes mellitus. We recently described marked ultrastructural (US) remodeling of the neurovascular unit (NVU) in type 2 diabetic *db*/*db* female mice. Herein, we tested whether the SGLT-2 inhibitor, empagliflozin (EMPA), protects the NVU from abnormal remodeling in cortical gray and subcortical white matter. Ten-week-old female wild-type and *db*/*db* mice were divided into lean controls (CKC, *n* = 3), untreated *db*/*db* (DBC, *n* = 3), and EMPA-treated *db*/*db* (DBE, *n* = 3). Empagliflozin was added to mouse chow to deliver 10 mg kg^−1^ day^−1^ and fed for ten weeks, initiated at 10 weeks of age. Brains from 20-week-old mice were immediately immersion fixed for transmission electron microscopic study. Compared to CKC, DBC exhibited US abnormalities characterized by mural endothelial cell tight and adherens junction attenuation and/or loss, pericyte attenuation and/or loss, basement membrane thickening, glia astrocyte activation with detachment and retraction from mural cells, microglia cell activation with aberrant mitochondria, and oligodendrocyte–myelin splitting, disarray, and axonal collapse. We conclude that these abnormalities in the NVU were prevented in DBE. Empagliflozin may provide neuroprotection in the diabetic brain.

## 1. Introduction

Type 2 diabetes mellitus (T2DM) is a chronic metabolic–endocrine disorder characterized by hyperglycemia, insulin resistance or relative lack of insulin, and this glucotoxic state promotes tissue oxidative/nitrosative stress and chronic inflammation. Importantly, T2DM is an independent risk factor for microvascular complications, which include neuropathy, retinopathy, nephropathy, diabetic cognopathy, and age-related neurodegenerative diseases such Alzheimer’s disease (AD) [1,2,3,4,5,6,7]. In this regard, we and others reported cognitive impairment, brain tissue oxidative stress, and ultrastructural (US) remodeling within the neurovascular unit (NVU) of cerebral cortical gray matter and transitional subcortical white matter from *db*/*db* mice relative to non-diabetic wild-type age- and gender-matched mice on the same background [1,2,3,8]. This preclinical model harbors many cardiovascular (macro- or microvascular), renal, and metabolic phenotypes of metabolic syndrome and T2DM in common with those in humans. Thus, as has been suggested previously [1,2,3], the *db*/*db* model could be instrumental in helping to elucidate whether current anti-hyperglycemic therapies, including sodium–glucose transporter-2 (SGLT-2) inhibitors, which lower serum glucose by inhibiting SGLT2-mediated glucose reabsorption in renal proximal tubules, could be neuroprotective. In this regard, it was recently reported that the SGLT-2 inhibitor, empagliflozin (EMPA), improved cognitive function and ameliorated oxidative stress in the brains of 17-week-old *db*/*db* mice [8]. Whether empagliflozin (EMPA) could prevent aberrant-maladaptive US remodeling in the NVU known to be present in 20-week-old female *db*/*db* mice [1,2,3] is unknown. In this investigation, we hypothesized that administration of EMPA for ten weeks might prevent or ameliorate pathological US remodeling of the NVU. Our observations demonstrate that EMPA protects the NVU in the setting of severe T2DM (Figure 1).

## 2. Methods

### 2.1. Animal Studies

All animal studies were approved by the Institutional Animal Care and Use Committees at the Harry S Truman Memorial Veterans’ Hospital and University of Missouri, Columbia, MO, USA (No.190), and conformed to the Guide for the Care and Use of Laboratory Animals published by the National Institutes of Health (NIH). Eight-week-old female *db*/*db* (BKS.Cg-*Dock7*^m^+/+*Lepr^db^*/J; DBC) and wild-type control (C57BLKS/J; CKC) mice were purchased from the Jackson Laboratory (Ann Harbor, MI, USA) and were housed under standard laboratory conditions where room temperature was 21–22 °C, and light and dark cycles were 12 h each. Three cohorts of mice were used: lean non-diabetic controls (CKC, *n* = 3), obese, insulin-resistant and diabetic *db*/*db* (DBC, *n* = 3), and *db*/*db* mice treated with the SGLT2 inhibitor, EMPA, to deliver 10 mg kg^−1^ day^−1^ and fed for 10 weeks, initiated at 10 weeks of age (DBE, *n* = 3). All mice were sacrificed for study at 20 weeks of age. We chose female mice because we have extensively characterized the cardiovascular phenotype, as well as US remodeling in the cortical NVU in female *db*/*db* mice [1,2,3,9,10]. Furthermore, women exhibit higher incidences of dementia/AD compared to men (especially in older age groups) [11].

### 2.2. Tissue Collection and Preparation for Transmission Electron Microscopy

The left hemispheres of brains were collected immediately upon sacrifice (Figure 2) and placed in a standard transmission electronic microscopy (TEM) fixative of 2% paraformaldehyde and 2% glutaraldehyde in 100 mM of sodium cacodylate buffer (pH = 7.35) for immersion fixation. Approximately 1-mm sections from the mid-cortical gray matter tissue were then rinsed with 100 mM sodium cacodylate buffer (pH 7.35) containing 130 mM sucrose. Secondary fixation was performed using 1% osmium tetroxide (Ted Pella, Inc., Redding, CA, USA) in cacodylate buffer using a Pelco Biowave (Ted Pella) operated at 100 W for 1 min. Specimens were next incubated at 4 °C for 1 h, then rinsed with cacodylate buffer, and further rinsed with distilled water. En bloc staining was performed using 1% aqueous uranyl acetate and incubated at 4 °C overnight, then rinsed with distilled water. Using the Pelco Biowave, a graded dehydration series (e.g., 100 W for 40 s) was performed using ethanol, transitioned into acetone, and dehydrated tissues were then infiltrated with Epon resin (250 W for 3 min) and polymerized at 60 °C overnight. Ultrathin sections were cut to a thickness of 85 nm using an ultramicrotome (Ultracut UCT, Leica Microsystems, Wetzlar, Germany) and stained using Sato’s triple lead solution stain and 5% aqueous uranyl acetate. Multiple images were acquired for study at various magnifications with a JOEL 1400-EX TEM JEOL (JEOL, Peabody, MA, USA) at 80 kV on a Gatan Ultrascan 1000 CCD (Gatan, Inc., Pleasanton, CA, USA).

### 2.3. Sample Preparation for Serial Block Face Imaging: Focused Ion Beam/Scanning Electron Microscopy for Supplemental Video

Samples were prepared following a modified version of NCMIR (National Center for Microscopy and Imaging Research) methods for three-dimensional (3D) electron microscopy. Unless otherwise stated, all reagents were purchased from Electron Microscopy Sciences and all specimen preparation was performed at the Electron Microscopy Core Facility, University of Missouri. Tissues were fixed in 2% paraformaldehyde, 2% glutaraldehyde in 100 mM sodium cacodylate buffer pH = 7.35. Next, fixed tissues were rinsed with 100 mM sodium cacodylate buffer, pH = 7.35 containing 130 mM sucrose. Secondary fixation was performed using equal parts 4% osmium tetroxide and 3% potassium ferrocyanide in cacodylate buffer and incubated on ice for 1 h, then rinsed with cacodylate buffer and further with distilled water. En block staining was performed for one hour in a 1% thiocarbohydrazide solution followed by distilled water rinses. Rinsed tissues were incubated in an additional 2% aqueous osmium tetroxide solution for 30 min at room temperature, then rinsed with distilled water. Additional en bloc staining was performed using 1% aqueous uranyl acetate and incubated at 4 °C overnight, then rinsed with distilled water. A final en bloc staining was performed using Walton’s Lead Nitrate solution (Lawrence Livermore Laboratory University of California, Livermore, CA, USA) for 30 min at 60 °C. Tissues were rinsed and dehydrated using ethanol, transitioned into acetone, and then infiltrated with Durcapan resin and polymerized at 60 °C overnight. Block faces were prepared using an ultramicrotome (Ultracut UCT, Leica Microsystems, Germany) and a diamond knife (Diatome, Hatfield, PA, USA) and mounted on an SEM stub and coated with 20 nm of platinum using the EMS 150T-ES Sputter Coater (Leica Microsystems Inc. Buffalo Grove, IL, USA). Serial block face data were acquired on a Thermo Fisher Scientific Scios Analytical Dualbeam (Hillsboro, OR, USA). The region of interest was identified using established landmarks and protected with a 1-μm layer of platinum using the ion column. Trenches were rough cut to the side and the front of the block face using a high ion beam current (30 kV 5 nA) to expose the desired block face. Next, the block face was polished using an ion beam current of 50 pA prior to collecting serial images using the Slice &View automated software package. Serial sections were cut at a thickness of 20 nm (30 kV 1 nA) and SEM images were acquired at 2 keV and 25 pA using the T1-BSE detector and reverse contrast. Image segmentation was performed using ThermoScientific Amira 6 Software (Thermo Fisher Scientific). Approximately 250 slices were obtained to create the video for Appendix A.

The regions of interest within each ultrathin section were selected based on the presence of NVU capillaries and myelin, structures that are readily identifiable within the cortical layer III utilizing TEM. Cortical gray matter layer III is identifiable due to the large number of pyramidal neuronal nuclei and the transitional zone subcortical white matter resides immediately beneath the cortical gray matter layers (I–VI). Images of ten NVU capillaries were taken from each mouse, thus providing at total of 30 each for CKC, DBC, and DBE (90 NVUs were examined in total). Eighty percent of the NVUs of diabetic DBC (24 of 30) exhibited maladaptive remodeling characterized by attenuation and/or loss of endothelial tight and adherens junctions, thickening of basement membranes (BMs), attenuation and/or loss of pericytes, astrocyte detachment or retraction, and dysmyelination and myelin disarray with axonal collapse. All of the NVUs and myelin from CKC and DBE appeared to have normal ultrastructural morphology.

It is appropriate to begin with the NVU images at lower magnifications in order to place them within the context of their surrounding proximal cellular elements within layer III of the cortical gray matter of the brain (Figure 1). Additionally, it is important to note that the abnormalities observed in DBC (Figure 1D) were not present in the NVU of mice treated with EMPA (DBE) (Figure 1E).

Basement membrane (BM) thickening is a constant finding in diabetic microvascular end-organ damage, which now includes the preclinical diabetic DBC brain [1]. Treatment with EMPA protected the BM from thickening, pericyte attenuation, and/or loss, detachment, and retraction of ACs from the NVU and attenuation and/or loss of EC blood–brain barrier (BBB) tight and adherens junctions (Figure 1 and Figure 3).

We frequently observed invasion by amoeboid—activated microglial cells (aMGCs) of the NVU in DBC. Importantly, we observed the NVUs were protected from not only activation of MGCs but also their invasive tendency since they were ramified and not activated in DBE and appeared similar to CKC (Figure 4).

We have previously demonstrated that there were aberrant mitochondria (aMt), which may be characterized by marked enlargement, loss of mitochondrial matrix electron density, and loss of crista in aMGCs and in other NVU cells including endothelial cell(s) (ECs), pericytes and their foot processes (Pc–PcP), astrocyte(s) (ACs), oligodendrocyte(s) (OLs), and myelinated and unmyelinated axons in DBC compared to CKC (Figure 5) [2]. There were occasional but infrequent aMt found in some non-activated MGCs in DBE that were also observed in CKC; however, aMt were not observed in DBE in any of the cells depicted in Figure 6.

Empagliflozin treatment protected cellular aMt formation in DBE (Figure 6) compared to DBC (Figure 5).

Cortical gray matter myelin US remodeling with splitting, separation, and aberrant Mt were ameliorated with EMPA (Figure 7).

Myelin disarray was a frequent observation in the transition zone subcortical white matter in DBC compared to CKC and DBE (Figure 8).

Multiple collapse of axons within myelinated neurons of the DBC were observed in the transitional zone, which was prevented in DBE (Figure 9).

Myelin disarray, splitting, and axonal collapse were present in DBC models, but not in CKC. Importantly, US remodeling was not observed in DBE (Figure 10).

Fixed images from the focused ion-beam scanning electron microscope (FIB/SEM) (Appendix A) illustrate the normal appearance of myelin without splitting, separation or disarray and normal appearing electron-dense mitochondria from the cortical gray matter in layer III in DBE (Figure 11).

There are marked differences in the morphology of DBC aberrant Mt within microglia cell and other glia and mural cells (EC and Pc) in capillary NVUs (Figure 5). Also, there appears to be no aMt and the myelin remains intact without splitting or separation in the EMPA treated models. We have made a direct side-by-side comparison on the pre-run fixed images for FIB/SEM between DBC untreated and treated DBC with EMPA-treated DBE (Figure 12).

Additionally, we have created a video of approximately 250 stacked images obtained with FIB/SEM technology to demonstrate that EMPA treatment (10–20 weeks) ameliorated aberrant US remodeling (Figure 13) and (Appendix A).

## 3. Discussion

Numerous reports have documented multiple behavioral cognitive impairments in *db*/*db* mice [8,12,13,14,15,16,17,18]. The presence of diabetic cognopathy (impaired cognition), in part, provided the rationale for use of this model to study the potential therapeutic effects of antihyperglycemic therapy on the underlying structural abnormalities associated with diabetic cognopathy. Importantly, EMPA provides an insulin-independent blood glucose lowering effect via its inhibition of glucose reabsorption in the proximal tubules of the kidney with subsequent increased urinary glucose excretion. Glycemic improvements as a result of EMPA treatment have been demonstrated to ameliorate cognitive impairment, attenuate oxidative stress and inflammation, and significantly increase brain-derived neurotrophic factor (BDNF) in cerebral tissues of T2DM *db*/*db* mice [8]. Also, glycemic improvements by EMPA were associated with reductions in vascular oxidative stress, advanced glycation end-products (AGE) and their receptor (RAGE) interaction, low-grade chronic inflammation and vascular dysfunction in the rat ZDF-Lepr (*fa/fa*) (leptin receptor deficient) rodent model of diabetes [19]. Furthermore, EMPA may have additional pleiotropic effects such as osmotic diuresis, increasing glucagon levels or yet to be discovered mechanisms to help understand its protective mechanisms in the DBC [19].

Type 2 diabetes mellitus is a polygenic, multifactorial, metabolic–endocrine disorder and complicated chronic disease characterized by obesity, insulin resistance, and hyperglycemia. The resulting glucotoxic state in female diabetic *db*/*db* mice promotes oxidative/nitrosative stress and chronic inflammation, which is associated with multiple diabetic microvascular end-organ complications [8,20,21]. In this regard, we recently reported cortical gray matter NVU, neuroglia, and myelin injury with US remodeling [1,2,3]. Glucotoxicity is considered a central mechanism contributing to brain injury involving the microvascular NVU and its composite mural and neuroglia cells [8]. Additionally, we were able to demonstrate that EMPA protected the NVU and its constituent cells (Figure 2, Figure 3, Figure 4, Figure 5 and Figure 6) and myelin (Figure 7, Figure 8, Figure 9, Figure 10 and Figure 11) from aberrant US remodeling. These aberrant US remodeling changes consisted of the following: (i) attenuation and/or loss of EC tight and adherent junctions of the BBB; (ii) activation of MGCs (amoeboid phenotype with marked increase in aberrant Mt and increased nuclear chromatin condensation) with invasion of the NVU; (iii) Pc attenuation and or loss; (iv) BM thickening; (v) aberrant Mt in ECs, MGCs, AC, Pc, myelinated/unmyelinated axons and OL (Figure 4 and Figure 5); (vi) myelin splitting–separation in cortical gray matter and myelin disarray in the transitional zone of the immediate subcortical white matter (Figure 7, Figure 8, Figure 9, Figure 10 and Figure 11; Appendix A).

Based on the observations reported in this study indicating neuroprotection by EMPA and the well characterized metabolic, cardiovascular, and brain ultrastructural phenotypes, we propose the following possible mechanisms to explain the aberrant remodeling changes we observed in the DBC and protection by EMPA. Importantly, severe hyperglycemia is well documented in female and male diabetic DBC, and EMPA has been shown to reduce glucose and HbA1c [8,9,10,19].

Importantly, empagliflozin has previously been demonstrated to lower blood glucose and HbA1c. We and others have demonstrated that the mean glucose levels were in the range of 100 mg/dL in the heterozygous db/+ (CKC) models; 500 mg/dL for the homozygous *db*/*db* (DBC) male and female models; and in the 200 mg/dL range in the EMPA treated models of the *db*/*db* mice with similar findings regarding the glucose lowering effect in the ZDF-Lepr (*fa*/*fa*) rat model [8,9,10,19]. Additionally, the duration of empagliflozin treatment paradigms were similar in the male *db*/*db* of 10 weeks (same as our paradigm) [8], the female *db*/*db* models of 5 weeks treatment [9,10], and the male ZDF-Lepr (*fa*/*fa*) models of 6 weeks treatment [19]. Furthermore, body weights remained comparable with slight increase in these above experiments and blood pressure remained comparable without any decrease in blood pressure [8,9,10]. Our current data regarding empagliflozin glucose lowering, body weights, and blood pressure for this experiment are comparable to the paper by Lin et al. (unpublished data in advanced preparation) [8].

As such, we propose that hyperglycemia-glucotoxicity associated with diabetes in the *db*/*db* female promotes the glucose-oxidative stress pathway via increased generation of reactive oxygen–nitrogen species (ROS/RNS) [3,8,9,10,12,22] (Figure 14). On the other hand, there are multiple situations that may, in part, contribute to the ultrastructural remodeling mechanisms from clinical scenarios as follows: aging, lifestyle, environment, genetics—particularly the leptin receptor deficiencies in the DBC and also the potentially numerous single-nucleotide polymorphisms (SNPs) in humans, and comorbidities associated with obesity (metabolic syndrome and accelerated atherosclerosis) [3].

The following limitations apply to this current study. First, we fixed tissue by immediate immersion rather than by perfusion fixation. We understand and appreciate that perfusion fixation is the “gold standard” for transmission electron microscopic ultrastructure studies. However, at this time, we prefer this immediate immersion fixation method as it may be comparable to the way in which human brain tissue is fixed following biopsy or acquisition following autopsied specimens. Second, this study was limited to the cortical gray matter (primarily layer III) and may not apply to other regions of the brain such as the hippocampus, mid-brain, or cerebellum. Third, this study was primarily designed and directed to interrogate the NVU capillaries and their immediate surroundings. Fourth, we utilized the female model of the *db*/*db* mouse and invariably the role of the female estrous cycle arises, which of course could have potential effects on the neurovascular structure and function. However, in this study the female model was chosen because females have greater impairments in diastolic relaxation and increased aortic stiffness and may predict future cardiovascular disease events [1,9]. In the future it would be interesting to compare the treatment results to other classes of glucose lowering anti-diabetic treatment modalities such as commonly first-line and most common treatment modality, metformin. Finally, this study was limited to only US observational findings and was not supported by functional studies such as NVU-coupling or functional hyperemia, immunohistochemistry, light microscopy or protein Western blots. As with any transmission electron microscopic study, the regions studied were very small (nano-micrometer) and reflect the very nature of TEM studies in general [1,2,3].

In summary, this investigation provides insight into the ultrastructural disease process of the diabetic brain in a preclinical model of diabetic cognopathy. Specifically, EMPA was effective in protecting aberrant ultrastructural remodeling in the NVU including the neuroglia. Future preclinical and clinical investigations are warranted to test whether EMPA affords neuroprotection specifically in the setting of T2DM. In this regard, results of the ongoing clinical trial entitled “Effect of empagliflozin versus placebo on brain insulin sensitivity in patients with prediabetes” (clinical trials.gov: (https://clinicaltrials.gov/ct2/show/study/NCT03227484#contacts) will be especially enlightening.

## 4. Conclusions

In conclusion, EMPA treatment (10–20 weeks) has demonstrated protection of the aberrant US remodeling phenotype in a study utilizing TEM and FIB/SEM technology. These findings suggest neuroprotection by EMPA via protection of the NVU, blood–brain barrier, US morphology of constituent NVU mural EC, and Pcs/neuroglia AC and MGCs and oligodendrocyte/myelin remodeling in the obese, insulin resistant female *db*/*db* mouse model of T2DM. Furthermore, these findings may subsequently offer neuroprotection from other age-related neurodegenerative diseases such as Alzheimer’s disease and Parkinson’s disease with impaired cognition. Indeed, these are exciting times to study the NVU and neuroglial ultrastructural morphology and observe the aberrant remodeling changes associated with the hyperglycemic female *db*/*db* model of T2DM, which were observed to be protected with EMPA treatment for ten weeks. We hope that our observed ultrastructural remodeling to the NVU and neuroglia in the DBC may currently provide or at some time in the future possess some translational value for others since EMPA treatment has been able to demonstrate a rather remarkable prevention of these remodeling abnormalities [23].

## Figures and Tables

**Figure 1 brainsci-09-00057-f001:**
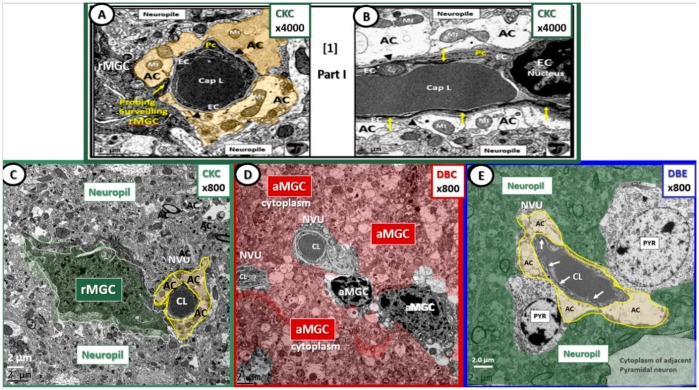
The neurovascular unit (NVU) in lean control (panels (**A**–**C**): CKC), untreated diabetic (panel (**D**): DBC), and diabetic mice treated with empagliflozin (EMPA) (panel (**E**): DBE). Panels (**A**) and (**B**) illustrate the normal ultrastructural morphology of the (NVU) in control CKC at higher magnification in order to demonstrate with greater clarity the contents of each cell comprising the NVU. Magnification 4000×; scale bar = 1 µm. Modified with permission from Reference [1]. Panel (**C**) illustrates a probing ramified microglia cell (rMGC) (pseudo-colored green) probing the NVU with an intact (pseudo-colored golden) halo or corona of astrocytes (ACs) within the confines of the neuropil. Panel (**D**) depicts an invasive (pseudo-colored red) activated microglia cell (aMGC) that has completely engulfed the NVU (uncolored) with markedly thickened basement membranes in DBC. Also, note the increased electron density and volume of chromatin within the aMGC nuclei in this image in addition to the detachment and retraction of ACs of the NVU. It is also important to note that there is a loss of pericytes and the normal intact ACs to form the halo—corona as in CKC and DBE (panels (**C**) and (**E**)). Panel (**E**) depicts an intact NVU, which is in close contact to two adjacent pyramidal (Pry) neurons in DBE and note the intact (pseudo-colored golden) AC halo—corona enveloping the endothelial cells similar to CKC controls with intact tight and adherens junctions within the endothelial cells (ECs) (arrows). Magnification 800×; scale bar = 2 µm (panels (**C**–**E**)). Figure images throughout text are color-coded with control CKC images outlined in green; diabetic *db*/*db* DBC in red; EMPA (SGLT2 inhibitor) treated DBE in blue in order to readily assist the reader in identification of each cohort. Cap = capillary; CL = capillary lumen.

**Figure 2 brainsci-09-00057-f002:**
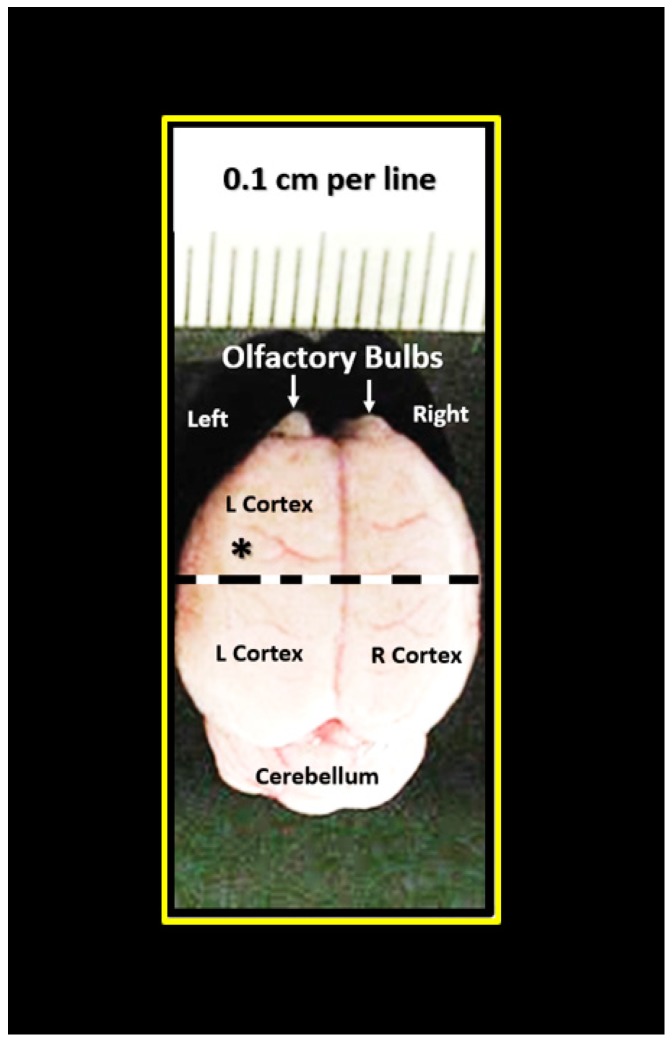
Photomicrograph of brain specimen for transmission electron microscopic studies. The left hemisphere was utilized for this study. Cortical gray matter tissue specimens were obtained just cephalad to the mid-cortex dashed line (asterisk). Scale = 0.1 cm. Brain measured approximately 1.4 × 1.5 cm upon removal.

**Figure 3 brainsci-09-00057-f003:**
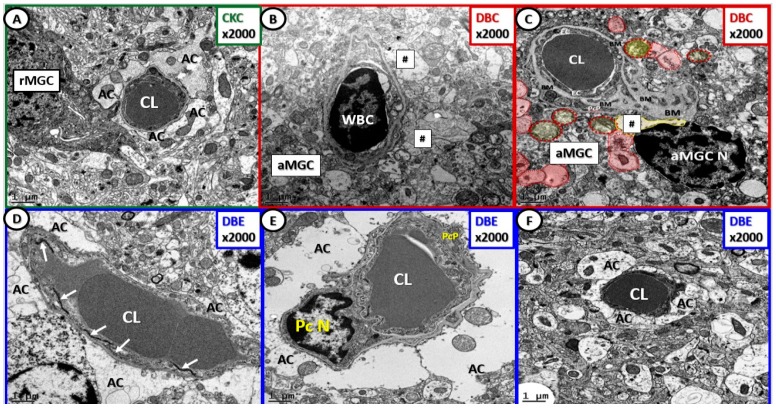
Empagliflozin protects the NVU from basement membrane (BM) thickening, loss of blood–brain barrier endothelial tight and adherens junctions and pericytes. Panel (**A**) depicts the normal ultrastructure morphologic phenotype of the NVU in the CKC. Panels (**B**) and (**C**) depict thickening of the NVU basement membrane, activation of NVU endothelial cells (white blood cell (WBC)—lymphocyte adhesion) with loss of blood–brain barrier tight and adherens junctions, inflammation and activation of endothelial cell in panel (**B**), and detachment and retraction of astrocytes (AC—hashtags and red pseudo-colored astrocytes) in panels (**B**) and (**C**). Panels (**D**), (**E**), and (**F**) depict the intact NVUs with protection from NVU cellular remodeling in DBE, and note they are similar to CKC (panel (**A**)). Note the intact TJ/AJ in panel (**D**) (arrows), the intact pericyte nucleus and pericyte process (PcP) in panel (**E**), and the intact astrocytes in panels (**D**), (**E**), and (**F**). At this magnification it is difficult to visualize the endothelial cell(s) that form the inner lining of the NVU capillary lumen (CL). Magnification 2000×; scale bar = 1 µm. AC = astrocyte end feet; aMGC = activated amoeboid microglial cell; rMGC = ramified microglial cell; CL = capillary lumen; PcN = pericyte nucleus; PcP = pericyte process.

**Figure 4 brainsci-09-00057-f004:**
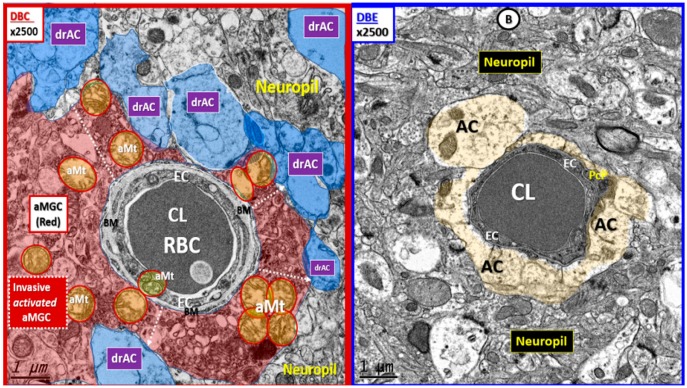
Basement membrane thickening, activated microglia invasion, and detachment of astrocyte foot processes are protected with empagliflozin. Panel (**A**) depicts the invasion of the activated MGC (aMGC) to totally engulf the NVU with detachment of AC end-foot processes (pseudo-colored blue; drAC = detached retracted astrocyte). Also, note the aberrant mitochondria (aMt) (pseudo-colored yellow with red outline) in DBC, which may be responsible for reactive oxygen species (ROS) production and leakage. Dashed arrows depict the detachment and retraction of the AC associated with aMGC invasion of the NVU. Panel (**B**) illustrates how empagliflozin protects the NVU from aberrant remodeling in DBE. Note the intact AC end feet (pseudo-colored golden) and that the mitochondria in DBE are electron dense and not aberrant as in the DBC. Panel (**B**) is a higher magnification of panel F in Figure 3. Magnification 2500×; scale bar = 1 µm. aMGC = activated microglia cell; aMt = aberrant mitochondria; drAC = detached retracted AC; EC = endothelial cell; CL = NVU capillary lumen; Pc = pericyte foot process; RBC = red blood cell.

**Figure 5 brainsci-09-00057-f005:**
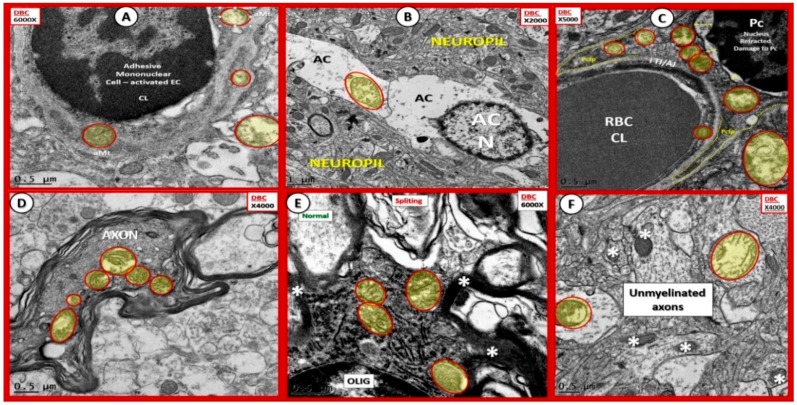
Aberrant mitochondria in endothelial cells, pericytes, and foot processes, astrocytes, oligodendrocytes, myelinated and unmyelinated neurons. Panels (**A**–**F**) demonstrate that aberrant mitochondria (aMt) were found to be present in multiple cells in addition to activated microglia cells (aMGCs). The aMt are pseudo-colored in each of these panels (yellow outlined in red lines) in order to allow their rapid recognition. Panels depict the aMt characterized by swollen mitochondria (Mt), loss of electron-dense Mt matrix and crista. Panel (**A**) illustrates the aMt within the endothelial cells and surrounding aMGC. Panel (**B**) depicts aMt within AC cytoplasm. Panel (**C**) demonstrates aMt in pericytes and foot processes (Pc and Pcfp). Panel (**D**) depicts aMt in a dysmyelinated neuronal axon. Panel (**E**) depicts aMt within an oligodendrocyte cytoplasm and Panel (**F**) illustrates aMt in an AC to the left and an unmyelinated axon on the right within the neuropil. Scale bars = 0.5 μm in all images except for panel (**B**) which has a scale bar = 1 μm [2].

**Figure 6 brainsci-09-00057-f006:**
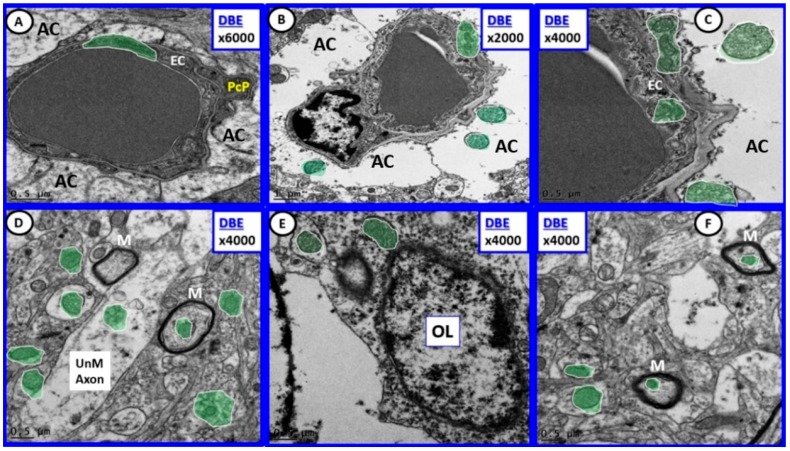
Diabetes-induced aberrant mitochondria remodeling was prevented by empagliflozin. Note that in all panels (**A**–**F**) the mitochondria are of smaller size and that the mitochondria matrix is electron dense (pseudo-colored green encircled with white lines) in contrast to the swollen aMt with loss of mitochondrial matrix electron density and loss of crista in the previous Figure 5. Magnification 6000× (panel (**A**)); 2000× (panel (**B**)), and 4000× (panels (**C**–**F**)); Scale bars = 0.5 μm in all images except for panel (**B**) which has a scale bar = 1 μm. AC = astrocyte; EC = endothelial cell; M = myelin; OL= oligodendrocyte; PcP = pericyte foot process; UnM = unmyelinated.

**Figure 7 brainsci-09-00057-f007:**
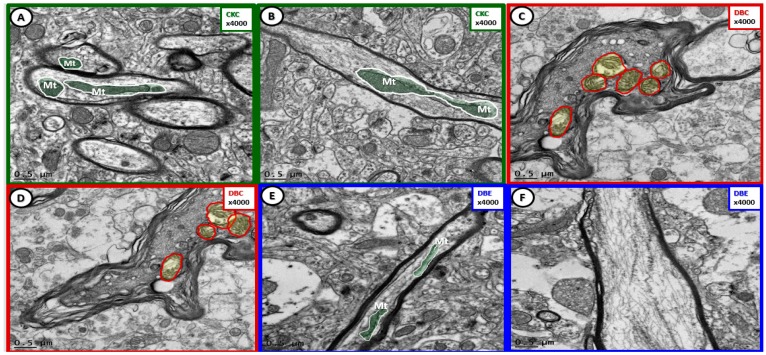
Cortical gray matter myelin remodeling was protected with empagliflozin treatment. Panels (**A**) and (**B**) demonstrate normal myelination and electron-dense mitochondria (Mt) in the cortical gray matter. Panels (**C**) and (**D**) depict marked myelin splitting and separation in DBC, and also note the aberrant mitochondria (aMt) within the axons (pseudo-colored yellow encircled by red outlines) as compared to the normal axonal mitochondria in CKC and DBC (pseudo-colored green with white outlines). Dysmyelination and aMt remodeling were prevented with empagliflozin treatment (panels (**E**), (**F**)). Magnification 4000×; scale bar = 0.5 µm.

**Figure 8 brainsci-09-00057-f008:**
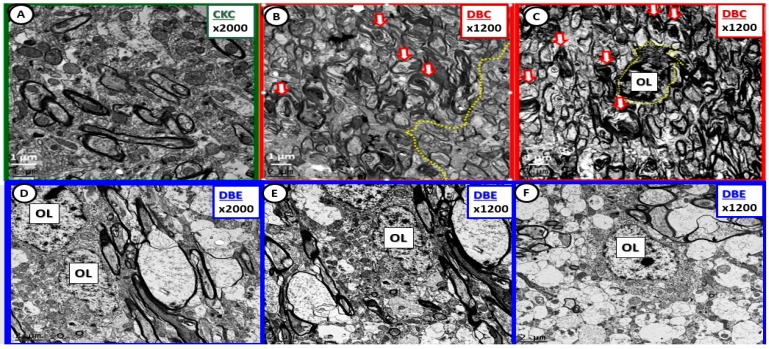
Diabetes-induced myelin disarray in the transitional zone was protected by empagliflozin treatment. Panel (**A**) illustrates the findings in the CKC. Panels (**B**) and (**C**) depict the marked myelin disarray, and note the more normal appearing myelin in panel (**B**) with the demarcation line (yellow dashed line) between more normal myelin and the myelin disarray. Also note the open arrows in panels (**B**) and (**C**), which depict axonal collapse within the myelinated axons. Panel (**C**) demonstrates an oligodendrocyte (OL) with increased chromatin condensation. In contrast note how treatment with empagliflozin results in less myelin disarray and illustrates normal oligodendrocyte morphology without chromatin condensation within the nucleus. Magnification 2000× in panels (**A**) and (**D**) with scale bar =1 µm. Magnification 1200×; scale bar = 1–2 µm (panels (**B**), (**C**), (**E**), (**F**)).

**Figure 9 brainsci-09-00057-f009:**
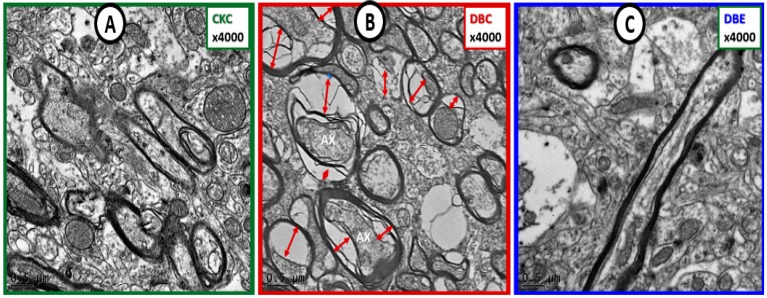
Diabetes-induced axonal collapse was prevented by empagliflozin. Panel (**A**) illustrates the normal morphology of myelinated axons and note the compactness of the myelin lamella. Panel (**B**) depicts multiple myelinated axons with collapsed axoplasm of axons (AX) (red double arrows) as compared to the myelinated neurons in control CKC (panel (**A**)). Panel (**C**) demonstrates that empagliflozin treatment protects axons from collapsing and appears similar to CKC in panel (**A**). Magnification 4000×; Bar = 0.5 µm. Ax = axon.

**Figure 10 brainsci-09-00057-f010:**
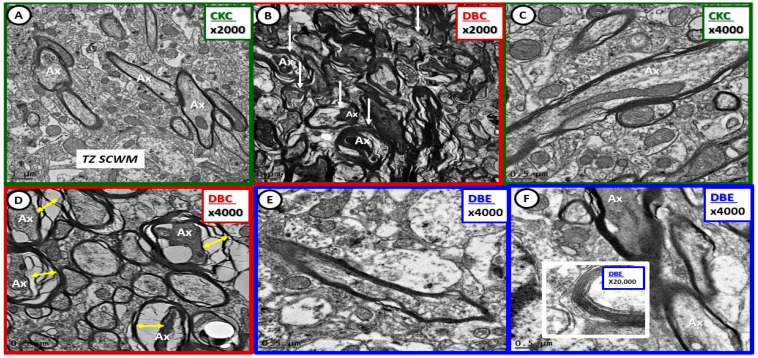
Diabetes-induced transitional zone subcortical white matter myelin and axonal remodeling was prevented with empagliflozin. Panels (**A**) and (**C**) demonstrate normal axon and myelin ultrastructural morphology in the transitional zone. Panels (**B**) and (**D**) depict myelin splitting, disarray, and axonal collapse, respectively (arrows and double arrows). Empagliflozin prevented the aberrant myelin and axonal remodeling (panels (**E**), (**F**)). Insert in panel (**F**) depicts the intact myelin lamellar units. Magnifications 2000×; scale bar = 1 µm (panels (**A**), (**B**)); magnifications 4000×; scale bar = 0.5 µm (panels (**C**)–(**F**)). Magnification 20,000×; scale bar = 100 nm (insert panel (**F**)). Ax = axon; TZ SCWM = transitional zone subcortical white matter.

**Figure 11 brainsci-09-00057-f011:**
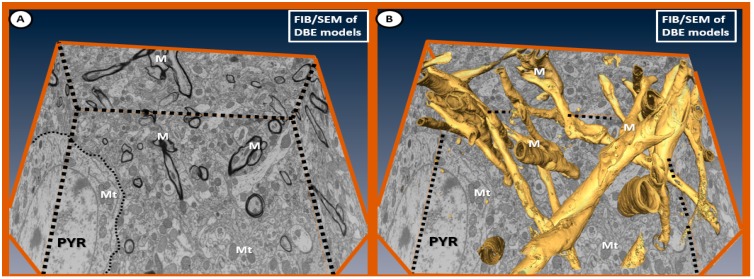
Fixed images of the focused ion-beam (FIB)/SEM emphasizing ultrastructural normal myelin in relation to the surrounding neuropil in DBE. Panel (**A**) depicts the markedly electron-dense myelin (M), and note that the mitochondria (Mt) are electron dense without any abnormal remodeling changes of aberrant Mt as found in DBC in prior images (especially Figure 3 and Figure 4). This 2D type of image with left and right sides folded up at the regions of the black-dashed lines illustrates the marked electron density of myelin allowing the computer to stack and create 3D images in panel (**B**). Note the large pyramidal (PYR) cell with electron-dense, normal appearing cytoplasmic mitochondria in both panels (**A**) and (**B**) within the cytoplasm of PYR cells. Importantly, there is no splitting, separation or disarray as noted in previously-fixed ultrastructural DBC images. Panel (**B**) illustrates the computerized developed 3D-type image of myelin as a result of stacking electron dense myelin, which allowed the computer program to pseudo-color myelin golden. Note the uniformity of the golden myelin without separation, splitting or disarray as compared to the previous DBC figure images.

**Figure 12 brainsci-09-00057-f012:**
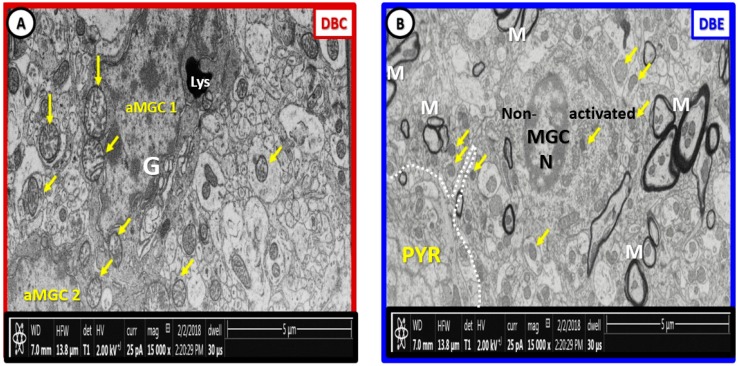
Labeled pre-run Appendix A fixed images comparing DBC and DBE. Panel (**A**) depicts the multiple enlarged more electron lucent aberrant mitochondria with loss of electron-dense mitochondria matrix and crista (arrows) in the diabetic DBC *db*/*db* model. In the empagliflozin-treated DBE model (panel (**B**)), note that the mitochondria are electron dense and smaller (arrows), which are morphologically similar in control non-diabetic lean models as in CKC in previously-fixed images (Figure 1A–C; Figure 3A–C; Figure 7A,B; Figure 8A; Figure 9A; Figure 10A,C) and that the myelinated axons are not split and separated as in previously depicted images and (Appendix A). Our observations support that the treatment with empagliflozin protects not only the aberrant Mt but also abnormal myelin remodeling in DBE models. Scale bar = 5 µm. aMGC = activated microglia cell; Lys = lysosome; M = myelin; MGC = microglia cell; N = nucleus; PYR = pyramidal neuron cell.

**Figure 13 brainsci-09-00057-f013:**
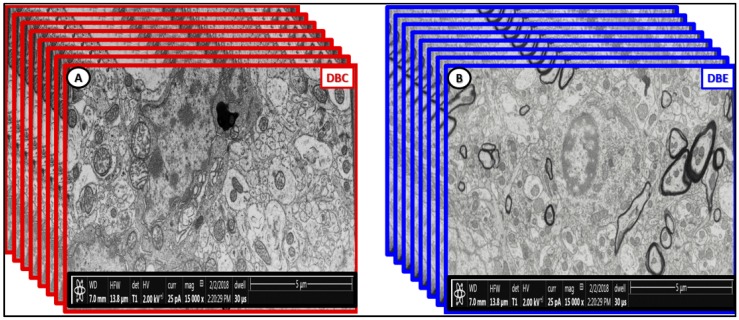
Stacked fixed images of diabetic DBC (outlined red) and EMPA-treated DBE models (outlined blue). The front pre-run images are labeled in Figure 12. The images in this figure are to demonstrate how Appendix A was created by utilizing approximately 250 stacked sliced images every 20 nm in order to create the Appendix A [2] and Appendix A. Scale bar = 5 µm.

**Figure 14 brainsci-09-00057-f014:**
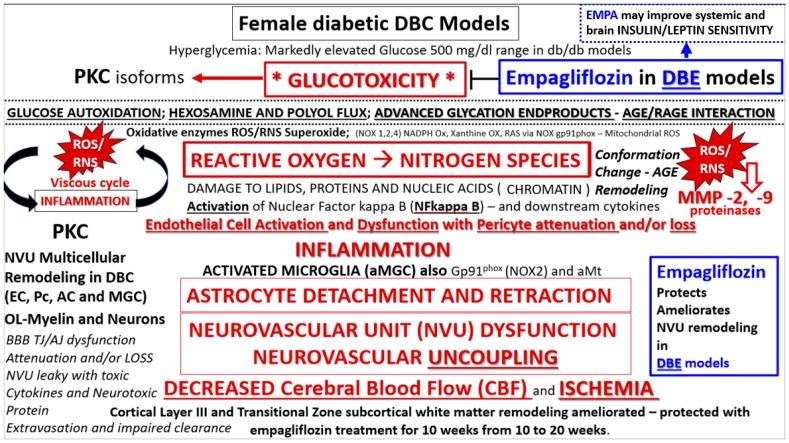
Possible mechanisms for remodeling changes in diabetic *db*/*db* mice and protection by EMPA. We propose an overarching hypothesis that hyperglycemia and glucotoxicity serve as a central injury mechanism to cells and tissues and are definitely upstream of diabetic end-organ complications, which may include microvascular disease and ultrastructure remodeling of the neurovascular unit in the brains of type 2 diabetes mellitus (T2DM) *db*/*db* mice and human patients. We have previously identified the observational abnormal remodeling associated with the NVU and its supportive cellular constituents (endothelial cell, pericyte, astrocyte, microglia and regional neurons) [1,2,3]. It is important to note that 10-week administration of EMPA protected the NVU and it supportive cells and neurons from US remodeling. In support of this, a recent paper showed that EMPA administration was associated with decreased cerebral-oxidative stress, increase brain-derived neurotrophic factor, and significantly prevented the impairment of cognitive function [8]. Importantly, note that empagliflozin in DBE models improves glucotoxicity and may also improve insulin and leptin sensitivity. AC = astrocyte; EC = endothelial cell; AGE = advance glycation end product; aMt = aberrant mitochondria; MGC = microglia cell; MMP = matrix metalloproteinase; NADPH Ox = nicotinamide adenine dinucleotide phosphate oxidase; *NOX*1,2,4 family genes encoding NADPH Ox 1,3,4 Ox = oxidase; PKC = protein kinase C; RAGE = receptor for AGE; RAS = renin angiotensin aldosterone system; RNS = reactive nitrogen species; ROS = reactive oxygen species; TJ/AJ = tight junction/adherens junction.

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
