# Peer review of "Empagliflozin Ameliorates Type 2 Diabetes-Induced Ultrastructural Remodeling of the Neurovascular Unit and Neuroglia in the Female db/db Mouse"

_brainsci, 2019, doi:10.3390/brainsci9030057_

Round 1
Reviewer 1 Report
In this study, the authors tested whether the SGLT-2 inhibitor, empagliflozin, protects the NVU from abnormal remodeling in cortical grey and subcortical white matter. They demonstrated the characteristics of abnormalities in the neurovascular unit in db/db mice and these abnormalities were prevented by empagliflozin.
The findings are some interesting and the study was well done. However, there are several concerns about this work to be addressed by the authors.
(1) The manuscript is limited to the description of phenomenological structural abnormalities in the neurovascular unit. Furthermore, discussion is limited to the description of structural abnormalities indicated by the authors and provides no deep insight.
(2) The reviewer is wondering whether the prevention of structural abnormalities in the neurovascular unit of type 2 diabetic mice is specific for SGLT2 inhibitor or not.
(3) Is the mechanism underlying the amelioration of structural abnormalities in the neurovascular unit attributed to only amelioration of hyperglycemia?
(4) Can empagliflozin penetrate blood brain barrier?
(5) Does SGLT2 exist in brain tissues such as neurovascular unit?
(6) In this study, empagliflozin was given to mice as chow containing 10 mg/kg/day. Is such a dose of empagliflozin appropriate for your experiment? Did the authors check blood glucose in each group of mice?
Author Response
RESPONSE TO REVIEWER #1
In this study, the authors tested whether the SGLT-2 inhibitor, empagliflozin, protects the NVU from abnormal remodeling in cortical grey and subcortical white matter. They demonstrated the characteristics of abnormalities in the neurovascular unit in db/db mice and these abnormalities were prevented by empagliflozin.
The findings are some interesting and the study was well done. However, there are several concerns about this work to be addressed by the authors.
(1) The manuscript is limited to the description of phenomenological structural abnormalities in the neurovascular unit. Furthermore, discussion is limited to the description of structural abnormalities indicated by the authors and provides no deep insight.
The insight to this manuscript is the finding that EMPA protects the NVU from the marked ultrastructural remodeling changes that have been demonstrated in previous publications (Parts I, II, III in Neuroglia) and also in this manuscript with empagliflozin as a treatment paradigm. While there is no deep insight other than the ultrastructural findings, we feel that these findings will aid others to examine in greater depth the functional insights associated with the ultrastructural remodeling changes. We regret that we could not do the functional studies in this experimental study at this time and so state in the limitations portion of the Discussion Section and further, we understand that the absence of functional studies are a limitation as follows:
Lines 380, 392-304: “ Finally, this study was limited to only US observational findings and was not supported by functional studies such as NVU-coupling or functional hyperemia, immunohistochemistry, light microscopy or protein Western blots.
Also we have attempted to strengthen the study by including blood glucose, body weights and blood pressure data by our group and others with a similar treatment paradigm. See discussion Section 4: Lines 342-353 of 4th paragraph.
4. Discussion
Importantly, empagliflozin has been previously demonstrated to lower blood glucose and HbA1c. We and others have demonstrated that the mean glucose levels were in the range of 100 mg/dl in the heterozygous db/+ (CKC) models; 500 mg/dl for the homozygous db/db (DBC) male and female models and 200 mg/dl range in the EMPA treated models of the db/db mice with similar findings regarding glucose lowering effect in the ZDF-Lepr(fa/fa) rat model [8-10,19]. Additionally, the duration of empagliflozin treatment paradigms were similar in the male db/db of 10 weeks (same as our paradigm) [8], the female db/db models of 5 weeks treatment [9, 10] and the male ZDF-Lepr(fa/fa) models of 6 weeks treatment [19]. Furthermore, body weights remained comparable with slight increase in these above experiments and blood pressure remained comparable without any decrease in blood pressure [8-10]. Our current data regarding empagliflozin glucose lowering, body weights and blood pressure for this experiment is comparable to that in paper by Lin B et. al. (unpublished data; submitted for publication) [8]
[8]. Lin B, Koibuchi N, Hasegawa Y, Sueta D, Toyama K, Uekawa K, Ma M, Nakagawa T, Kusaka H, Kim-Mitsuyama S: Glycemic control with empagliflozin, a novel selective SGLT2 inhibitor, ameliorates cardiovascular injury and cognitive dysfunction in obese and type 2 diabetic mice. Cardiovascular Diabetology. 2014, 13:148. doi: 10.1186/s12933-014-0148-1.
[9].Habibi J, Aroor AR, Sowers JR, Jia G, Hayden MR, Garro M, Barron B, Mayoux E, Rector RS, Whaley-Connell A, DeMarco VG: Sodium glucose transporter 2 (SGLT2) inhibition with empagliflozin improves cardiac diastolic function in a female rodent model of diabetes. Cardiovasc Diabetol. 2017 Jan 13;16(1):9. doi: 10.1186/s12933-016-0489-z.
[10]. Aroor AR, Das NA, Carpenter AJ, Habibi J, Jia G, Ramirez-Perez FI, Martinez-Lemus L, Manrique-Acevedo CM, Hayden MR, Duta C, Nistala R, Mayoux E, Padilla J, Chandrasekar B, DeMarco VG: Glycemic control by the SGLT2 inhibitor empagliflozin decreases aortic stiffness, renal resistivity index and kidney injury. Cardiovasc Diabetol. 2018 Jul 30;17(1):108. doi: 10.1186/s12933-018-0750-8.
[19]. Steven S, Oelze M, Hanf A, Kröller-Schön S, Kashani F, Roohani S, et. al.: The SGLT2 inhibitor empagliflozin improves the primary diabetic complications in ZDF rats. Redox Biol. 2017 Oct;13:370-385. doi: 10.1016/j.redox.2017.06.009.
(2) The reviewer is wondering whether the prevention of structural abnormalities in the neurovascular unit of type 2 diabetic mice is specific for SGLT2 inhibitor or not.
This is a great point brought up by the reviewer and we cannot answer this important question since we did not study the other glucose lowering treatment paradigms. However, if this protection of ultrastructure remodeling is due to the lowering of glucose (glucotoxicity), we would anticipate other glucose lowering treatments with other oral anti-diabetic medications would also help to protect the NVU ultrastructure remodeling changes as well. However, if for example metformin or other anti-diabetic treatments would lower glucose to similar levels but not affect or protect NVU from remodeling that would have made our empagliflozin treatment paradigm even more novel. For example, metformin, which is the most commonly prescribed therapy to treat hyperglycemia in T2DM, has been reported to improve cognitive function and abnormal glucose metabolism in patients with non-dementia vascular cognitive impairment [1] Thus, it is possible that other antihyperglycemic drugs, like metformin could similarly prevent the histopathology observed in our study. On the other hand, SGLT2 inhibitors exhibit pleiotropic effects beyond glucose control that could contribute to the microvascular protection reported for this group of drugs (see references 8-10,19 in comment 1).
[1] Lin Y, Wang K, Ma C, Wang X, Gong Z, Zhang R, et al. Evaluation of Metformin on Cognitive Improvement in Patients With Non-dementia Vascular Cognitive Impairment and Abnormal Glucose Metabolism. Front Aging Neurosci. 2018;10:227. https://doi.org/10.3389/fnagi.2018.00227
(3) Is the mechanism underlying the amelioration of structural abnormalities in the neurovascular unit attributed to only amelioration of hyperglycemia?
This is another great point by reviewer. Authors cannot be certain that the protective effects are only due to glucose lowering. We have anticipated that this is the primary overarching mechanism; however, as pointed out in the discussion there could be other mechanisms in play with EMPA… such as the comment in discussion as follows in lines 319-321: “Furthermore, EMPA may have additional pleiotropic effects such as osmotic diuresis, increasing glucagon levels or yet to be discovered mechanisms to help understand its protective mechanisms in the DBC [19].” Decreasing obesity or blood pressure could also have an effect; however, we know in this model that the body weights and blood pressures were not lowered in this 10 week treatment paradigm and also in the other treatment paradigms in these references.
Reference [19] specifically discusses this possibility and references [8] and [19] have utilized EMPA with positive findings, which both groups suggest it is the glucose lowering effect that is protective and results in their outcomes studies:
[8]. Lin B, Koibuchi N, Hasegawa Y, Sueta D, Toyama K, Uekawa K, Ma M, Nakagawa T, Kusaka H, Kim-Mitsuyama S: Glycemic control with empagliflozin, a novel selective SGLT2 inhibitor, ameliorates cardiovascular injury and cognitive dysfunction in obese and type 2 diabetic mice. Cardiovascular Diabetology. 2014, 13:148. doi: 10.1186/s12933-014-0148-1.
[19]. Steven S, Oelze M, Hanf A, Kröller-Schön S, Kashani F, Roohani S, et. al.: The SGLT2 inhibitor empagliflozin improves the primary diabetic complications in ZDF rats. Redox Biol. 2017 Oct;13:370-385. doi: 10.1016/j.redox.2017.06.009.
Specifically Lin B et. al. [8] demonstrated that the approximate mean blood glucose in control heterozygous dbM was around 100 mg/dl, the diseased homozygous db/db was 500 mg/dl and that that EMPA treated models was 200 mg/dl ranges.
Additionally, Steven S. et. al. [19] demonstrated in the ZDF-Lepr(fa/fa) rat model of T2DM the following glucose levels: Blood glucose was higher in all ZDF rats (more than 4-fold compared to the lean control rats) before SGLT2i treatment. Fasting blood glucose levels were decreased by approximately 40% after 6 weeks of SGLT2i treatment (both doses), with a trend for more pronounced ant-hyperglycemic effects of the higher dose SGLT2i treatment but were still higher than blood glucose levels in lean control rats. The parameter for long-term glycemic conditions, HbA1c, was elevated in ZDF rats and was significantly decreased by both doses of empagliflozin. Insulin resistance, measured by the HOMA-IR index, was significantly increased in the ZDF rats and the HOMA-IR index, was significantly increased in ZDF rats and was improved by the higher dose of empagliflozin. Accordingly, β-cell function as measured by the HOMA
beta index, was severely impaired in untreated ZDF rats and significantly improved by both doses of empagliflozin. These authors stated that Body weights and heart/body weight ratios were not significantly changed by SGLT2i treatment of ZDF rats.
“Importantly, we and others [8, 9, 19] have demonstrated that the mean glucose levels were in the 500 mg/dl range for the homozygous db/db (DBC) mice models and ZDF-Lepr(fa/fa) rat models; 100 mg/dl range in the heterozygous db/+ (CKC)models and 200 mg/dl range in the EMPA treated models of the db/db mice and ZDF-Lepr(fa/fa) rat models.
(4) Can empagliflozin penetrate blood brain barrier?
To our knowledge it is not currently known in the literature as to whether or not the SGLT2 inhibitors are able to penetrate the blood-brain barrier for certain. We feel it may possibly be able to do so and if it does this would help to increase our knowledge as to how it may offer such a protective nature in preventing the ultrastructural remodeling of the NVU. However, since others have demonstrated the protective role of glucose lowering in the prevention of impaired cognition with empagliflozin it may be due to glucose lowering in the periphery. Currently, authors are not certain as to the ability of empagliflozin to penetrate the blood-brain barrier and the answer remains illusive at this time.
However, even if it is not able to penetrate healthy blood-brain barriers we know from this study that the NVU and TJ/AJ have lost their integrity in some NVU of the cortex in layer III in the diabetic DBC models as compared to control CKC models and due to the loss of integrity of these NVU, empagliflozin may be able to enter these regions where the NVU has lost its integrity
I did find a comment regarding the inability of empagliflozin to penetrate the blood-brain barrier as follows:
“empagliflozin concentrations were absent included the brain, spinal cord, bone, bone marrow, eyes, eye lens, testis and uveal tract.
https://www.pharmacodia.com/yaodu/html/v1/chemicals/0987b8b338d6c90bbedd8631bc499221.html accessed 18 Feb2019.
(5) Does SGLT2 exist in brain tissues such as neurovascular unit?
It has been previously noted that there exists a SGLT1 receptor in the brain in rat models [*]; however. it may be suggested or hypothesized that the SGLT2 receptor is present or upregulated only in times of injury or response to injury. Interestingly, there has been recent validation of SGLTs (including specifically SGLT2) receptors identified in the male and female human brain tissue and this remains to be more fully elucidated in human diabetic models. The SGLT receptors were primarily located on neurons in the cortex and HC [***]. Of interest at least to me is also the recent demonstration that SGLT receptors on the microvascular cells within human astrocytoma’s indicating that this is at least a possibility for SGLTr to be present under some circumstances [****]
[*] Yu AS, Hirayama BA, Barrio JR: Functional expression of SGLTs in rat brain. Am J Physiol Cell Physiol. 2013;304(3):C240-47. doi: 10.1152/ajpcell.00317.2012.
Furthermore, SGLT1 receptors are located in the cortex [**]
[**] Yu AS1, Hirayama BA, Timbol G, Liu J, Diez-Sampedro A, Kepe V, Satyamurthy N, Huang SC, Wright EM, Barrio JR: Regional distribution of SGLT activity in rat brain in vivo. Am J Physiol Cell Physiol. 2013 Feb 1;304(3):C240-7. doi: 10.1152/ajpcell.00317.2012. Epub 2012 Nov 14.
[***] Oerter S, Förster C, Bohnert M: Validation of sodium/glucose cotransporter proteins in human brain as a potential marker for temporal narrowing of the trauma formation. Int J Legal Med. 2018; doi: 10.1007/s00414-018-1893-6. [Epub ahead of print]
[****] Wright EM, Ghezzi C Loo DF:Novel and Unexpected Functions of SGLTs. Physiology (Bethesda). 2017;32(6):435-443. doi: 10.1152/physiol.00021.2017.
(6) In this study, empagliflozin was given to mice as chow containing 10 mg/kg/day. Is such a dose of empagliflozin appropriate for your experiment? Did the authors check blood glucose in each group of mice?
Authors have previously utilized (60 mg kg−1 of diet) calculated to deliver 10 mg kg−1 day−1 based on food intake and this dose of EMPA significantly improves HbA1c, 2 h glucose concentration during oral glucose tolerance test (OGTT), insulin sensitivity by insulin euglycemic-hyperinsulinemic clamp and tends to lower circulating lipids in 17 week-old female db/db mice and that is why we used this dosage [*].
Others have used similar doses of empagliflozin with blood glucose lowering effects [8]. Authors did check blood glucose weights and blood pressure in each of these models; however, we are awaiting a submitted paper that discloses these results in the original study (unpublished data; submitted for publication pending). Once this is published, we will be able to share these results in this paper with permission. In the interim we have shared these results from a 5-10 week empagliflozin treated paradigm from Habibi J et al: Aroor A et. al. and Lin B et.al and Stevens S et.al along with the body weights, blood glucose, HbA1c and blood pressure which was not changed between the control CKC, the db/db DBC and db/db empagliflozin treated DBE models.

Reviewer 2 Report
General comments
This manuscript by Melvin R Hayden et al seeks to study the therapeutic effects of Empagliflozin on neurovascular unit remodeling in type 2 diabetic db/db female mice. The manuscript was well written in the language. However, the overall significance of this study is compromised due to lacking critical experiments to support the conclusion other than over-interpretation of many results. The concerns are described below:
Specific comments
1. The present study utilized female db/db mouse model but there was no characterization of essential baseline parameters including plasma glucose, HbA1, and body weight, etc.
2. Empagliflozin is a SGLT-2 inhibitor that reduces plasma glucose level to a normal range. The author should measure the plasma glucose and HbA1c level of the db/db mouse model after treatment of Empagliflozin to validate the efficacy of drug treatment in the present study.
3. Empagliflozin also reduces sodium reuptake that could lower blood pressure. In addition, previous reports indicated that SGLT2 inhibitors could lower body weight. Without examination of blood pressure and body weight, it is hard to conclude that the structural changes of NVC are due to Empagliflozin treatment.
4. Estrogen levels of female mice vary according to estrous cycles that may have potential effects on the neurovascular structure and function. The author should address whether they monitored the estrous cycle, and if not, why.
5. There are no functional studies of neurovascular-coupling such as functional hyperemia, thus it can not be concluded that all these structural changes play any role in altering NVC function.
6. Figure 4, what is the definition of aberrant mitochondria (aMt)? The authors labeled some aMt as ROS since these aMts are responsible for ROS production. How can you make this conclusion without any examination of ROS production vs. structure changes?
7. Figure 3D, what do the arrows point to? There are missing words in the figure legends.
Author Response
Response to Reviewer #2
This manuscript by Melvin R Hayden et al seeks to study the therapeutic effects of Empagliflozin on neurovascular unit remodeling in type 2 diabetic db/db female mice. The manuscript was well written in the language. However, the overall significance of this study is compromised due to lacking critical experiments to support the conclusion other than over-interpretation of many results. The concerns are described below:
Specific comments
1. The present study utilized female db/db mouse model but there was no characterization of essential baseline parameters including plasma glucose, HbA1, and body weight, etc.
We and others have previously reported that the dose of empagliflozin(10 mg/kg/day) used in our current study improves insulin sensitivity, HbA1c and fasting glucose in 16 week old female db/db mice [9, 10]. Moreover, this dose had no effect on body weight in either of these previous studies. Thus, it has been established that this dose of empagliflozin improves glycemia in our model without affecting body weight. In the revised submission we emphasize and have previously cited these two papers by Habibi J el.al and Aroor A et. al. [9, 10] is effective and improves glycemia. In the current study, we continued to use the same dose of empagliflozin and similarly observed that it lowered HbA1c from 11.0 +/- 0.06 to 5.9 +/= 0.2% (p<0.05); FG from 264 +/- 63 to 194 +/-47 mg/dl (p<0.05); and="" had="" no="" effect="" on="" body="" weight="" 56.2versus="" 59.4="" p="">0.05). Although we do not include these data in the current study, our intention is to report them in a manuscript (in advanced preparation) reporting on the efficacy of EMPA , along or in combination with Linagliptin, on cardiac and vascular function, a project which was the basis for the investigator-initiated proposal that was funded (DeMarco VG).
Also see response to comments (1-3) below that has been inserted in the 4. Discussion
2. Empagliflozin is a SGLT-2 inhibitor that reduces plasma glucose level to a normal range. The author should measure the plasma glucose and HbA1c level of the db/db mouse model after treatment of Empagliflozin to validate the efficacy of drug treatment in the present study.
See response to comments 1 and 3.
3. Empagliflozin also reduces sodium reuptake that could lower blood pressure. In addition, previous reports indicated that SGLT2 inhibitors could lower body weight. Without examination of blood pressure and body weight, it is hard to conclude that the structural changes of NVC are due to Empagliflozin treatment.
We previously reported a comprehensive assessment of ambulatory blood pressure (via constant radiotelemetric monitoring) in 16 week old female db/db mice compared to lean non-diabetic counterparts and db/db mice treated with the same dose of empagliflozin used in this study [9]. We were able to demonstrate that empagliflozin had no effect on systolic or diastolic blood pressure during either the light or dark cycle. The current study is a follow up study to our previous report.
[9].Habibi J, Aroor AR, Sowers JR, Jia G, Hayden MR, Garro M, Barron B, Mayoux E, Rector RS, Whaley-Connell A, DeMarco VG: Sodium glucose transporter 2 (SGLT2) inhibition with empagliflozin improves cardiac diastolic function in a female rodent model of diabetes. Cardiovasc Diabetol. 2017 Jan 13;16(1):9. doi: 10.1186/s12933-016-0489-z.
Authors have attempted to respond to reviewer’s comments (1-3) in the manuscript by inserting the additions within the 4 Discussion in the revised manuscript as follows:
See Discussion Section 4: Lines (342-353) of 4th paragraph. Please note that we inserted the revision of manuscript prior to our proposal of Possible Mechanisms (Figure 14).
4. Discussion
“Importantly, empagliflozin has been previously demonstrated to lower blood glucose and HbA1c. We and others have demonstrated that the mean glucose levels were in the range of 100 mg/dl in the heterozygous db/+ (CKC) models; 500 mg/dl for the homozygous db/db (DBC) male and female models and 200 mg/dl range in the EMPA treated models of the db/db mice with similar findings regarding glucose lowering effect in the ZDF-Lepr(fa/fa) male rat model [8-10,19]. Additionally, the duration of empagliflozin treatment paradigms were similar in the male db/db of 10 weeks (same as our paradigm) [8], the female db/db models of 5 weeks treatment [9, 10] and the male ZDF-Lepr(fa/fa) models of 6 weeks treatment [19]. Furthermore, body weights remained comparable with slight increase in these above experiments and blood pressure remained comparable without any decrease in blood pressure [8-10]. Our current data regarding empagliflozin glucose lowering, body weights and blood pressure for this experiment is comparable to that in paper by Lin B et. al. (unpublished data; submitted for publication) [8].”
[8]. Lin B, Koibuchi N, Hasegawa Y, Sueta D, Toyama K, Uekawa K, Ma M, Nakagawa T, Kusaka H, Kim-Mitsuyama S: Glycemic control with empagliflozin, a novel selective SGLT2 inhibitor, ameliorates cardiovascular injury and cognitive dysfunction in obese and type 2 diabetic mice. Cardiovascular Diabetology. 2014, 13:148. doi: 10.1186/s12933-014-0148-1.
[9].Habibi J, Aroor AR, Sowers JR, Jia G, Hayden MR, Garro M, Barron B, Mayoux E, Rector RS, Whaley-Connell A, DeMarco VG: Sodium glucose transporter 2 (SGLT2) inhibition with empagliflozin improves cardiac diastolic function in a female rodent model of diabetes. Cardiovasc Diabetol. 2017 Jan 13;16(1):9. doi: 10.1186/s12933-016-0489-z.
[10]. Aroor AR, Das NA, Carpenter AJ, Habibi J, Jia G, Ramirez-Perez FI, Martinez-Lemus L, Manrique-Acevedo CM, Hayden MR, Duta C, Nistala R, Mayoux E, Padilla J, Chandrasekar B, DeMarco VG: Glycemic control by the SGLT2 inhibitor empagliflozin decreases aortic stiffness, renal resistivity index and kidney injury. Cardiovasc Diabetol. 2018 Jul 30;17(1):108. doi: 10.1186/s12933-018-0750-8.
[19]. Steven S, Oelze M, Hanf A, Kröller-Schön S, Kashani F, Roohani S, et. al.: The SGLT2 inhibitor empagliflozin improves the primary diabetic complications in ZDF rats. Redox Biol. 2017 Oct;13:370-385. doi: 10.1016/j.redox.2017.06.009.
4. Estrogen levels of female mice vary according to estrous cycles that may have potential effects on the neurovascular structure and function. The author should address whether they monitored the estrous cycle, and if not, why.
We did not monitor the estrous cycle in this study and the following insertion with references was made in the 4. Discussion section lines 388-392: regarding limitations of this study as follows:
“Fourth, we utilized the female model of the db/db mouse and invariably the role of the female estrous cycle arises, which of course could have potential effects on the neurovascular structure and function. However, in this study the female model was chosen because females have greater impairments in diastolic relaxation and increased aortic stiffness and may predict future cardiovascular disease events [1], [9]”
5. There are no functional studies of neurovascular-coupling such as functional hyperemia, thus it can not be concluded that all these structural changes play any role in altering NVC function.
Indeed, the reviewer is correct in that we did not evaluate NVU coupling via functional hyperemia; however, we were able to demonstrate that the NVU ultrastructural phenotypic morphology is protected by empagliflozin treatment and therefore we can only assume that the functional hyperemia would be protected to a similar degree as it would be in the control CKC models. This was addressed in 4. Discussion section lines: 392-394 changes insertion in blue color as follows:
“Finally, this study was limited to only US observational findings and was not supported by functional studies such as NVU-coupling or functional hyperemia, immunohistochemistry, light microscopy or protein Western blots.”
6. Figure 4, what is the definition of aberrant mitochondria (aMt)? The authors labeled some aMt as ROS since these aMts are responsible for ROS production. How can you make this conclusion without any examination of ROS production vs. structure changes?
Aberrant mitochondria (aMt) are now defined in revised manuscript lines [193-197] as follows and changes are in blue color :
“We have previously demonstrated that there were aberrant mitochondria (aMt), which may be characterized by marked enlargement, loss of mitochondrial matrix electron density and loss of crista in aMGCs and in other NVU cells including endothelial cell(s) (EC), pericytes and their foot processes (Pc – PcP), astrocyte(s) (AC), oligodendrocyte(s) (OL) and myelinated and unmyelinated axons in DBC compared to CKC (Fig.5) [2].”
Author’s have removed the original Figure 4A and replaced it with new Figure 4 in revised manuscript. We have removed “ROS” from the original manuscript in Figure 4A. Also, we have revised the legend for this figure in lines [182-192] as follows: Note changes in blue color.
Figure 4. Basement membrane thickening, activated microglia invasion and detachment of astrocyte foot processes are protected with empagliflozin. Panel A depicts the invasion of the activated MGC (aMGC) to totally engulf the neurovascular unit (NVU} with detachment of AC end foot processes (pseudo-colored blue; drAC = detached retracted astrocyte). Also, note the aberrant mitochondria (aMt) (pseudo-colored yellow with red outline) in DBC, which may be responsible for reactive oxygen species (ROS) production and leakage. Dashed arrows depict the detachment and retraction of the AC associated with aMGC invasion of the NVU. Panel B illustrates how empagliflozin protects the NVU from aberrant remodeling in DBE. Note the intact AC end feet (pseudo-colored golden) and that the mitochondria in DBE are electron dense and not aberrant as in the DBC. Panel B is a higher magnification of panel F in figure 3. Magnification x2500; scale bar = 1 µm. aMGC = activated microglia cell; aMt = aberrant mitochondria; drAC = detached retracted AC EC = endothelial cell; CL = NVU capillary lumen; Pc = pericyte foot process; RBC = red blood cell.
7. Figure 3D, what do the arrows point to? There are missing words in the figure legends.
The arrows point to the endothelial tight and adherens junctions in figure 3D. This has been identified in the revised legend for figure 3 and the revised legend also have replaced the missing words as follows in the revised manuscript in lines [ 164-176] in blue lettering as follows:
Figure 3. Empagliflozin protects the NVU from basement membrane thickening, loss of blood-brain barrier endothelial tight and adherens junctions and pericytes. Panels A depicts the normal ultrastructure morphologic phenotype of the neurovascular unit (NVU) in the CKC. Panels B and C depict thickening of the NVU basement membrane, activation of NVU endothelial cells (white blood cell (WBC) - lymphocyte adhesion) with loss of blood-brain barrier tight and adherens junctions, inflammation and activation of endothelial cell in panel B and detachment and retraction of astrocytes (AC – hashtags and red pseudo-colored astrocytes) in panels B and C. Panels D, E and F depict the intact NVUs with protection from NVU cellular remodeling in DBE and note they are similar to CKC (panel A). Note the intact TJ/AJ panel D (arrows), the intact pericyte nucleus and pericyte process (PcP) in panel E and the intact astrocytes in panels D, E and F. At this magnification it is difficult to visualize the endothelial cell(s) that form the inner lining of the NVU capillary lumen (CL). Magnification x2000; scale bar = 1µm. AC = astrocyte end feet; aMGC = activated amoeboid microglial cell; rMGC = ramified microglial cell; CL = capillary lumen; Pc N = pericyte nucleus; PcP = pericyte process

Round 2
Reviewer 1 Report
This revised manuscript has sufficiently addressed the reviewer’s concerns.
Author Response
Authors would like to thank the reviewer's comments.
Reviewer 2 Report
The authors addressed all my concerns. I have no additional comments.
Author Response

(The authors gave the same response as above.)
